# Expressive rule-based modeling and fast simulation for dynamic compartments

**Till Köster[1], Philipp Henning [1,2]\*, Tom Warnke[1,3], Adelinde Uhrmacher[1]**

**1** Institute for Visual and Analytic Computing, University of Rostock, Rostock, Germany, **2** Institute of Medical Biochemistry and Molecular Biology, University Medicine Rostock, Rostock, Germany, **3** Limbus Medical Technologies GmbH, Rostock, Germany

\* philipp.henning@uni-rostock.de

## Abstract

Compartmentalization is vital for cell biological processes. The field of rule-based stochastic simulation has acknowledged this, and many tools and methods have capabilities for compartmentalization. However, mostly, this is limited to a static compartmental hierarchy and does not integrate compartmental changes. Integrating compartmental dynamics is challenging for the design of the modeling language and the simulation engine. The language should support a concise yet flexible modeling of compartmental dynamics. Our work is based on ML-Rules, a rule-based language for multi-level cell biological modeling that supports a wide variety of compartmental dynamics, whose syntax we slightly adapt. To develop an efficient simulation engine for compartmental dynamics, we combine specific data structures and new and existing algorithms and implement them in the Rust programming language. We evaluate the concept and implementation using two case studies from existing cell-biological models. The execution of these models outperforms previous simulations of ML-Rules by two orders of magnitude. Finally, we present a prototype of a WebAssembly-based implementation to allow for a low barrier of entry when exploring the language and associated models without the need for local installation.

**Data Availability Statement:** All codes used in this study is available at https://git.informatik.uni-rostock.de/mosi/ml-rules3-official.

## 1 Introduction

Compartmentalization is a central aspect of cell biological systems [1]. Modeling such systems as interrelated compartments in and between which processes occur allows approximating the system structure and its effect on the processes. To include the causal influence of compartments on the processes, it suffices to support *static compartments*, that is, compartments that do not change. However, to account for the causal influence of the occurring processes on the compartmental structure and interactions, the model must support *dynamic compartments*. This includes the creation of new compartments, their deletion, merging, and splitting of compartments, as well as one compartment entering or leaving another (Fig 1).

The reciprocal influence of compartments and the processes that occur in and between them can be observed in many cell biological systems. For example, the life cycle of a cell can be modeled as intracellular processes affecting cell growth, ultimately leading to cell division

**Funding:** P.H. received funding from the Deutsche Forschungsgemeinschaft (DFG, German Research Foundation, https://www.dfg.de/en) SFB 1270 – 299150580 ELAINE and the Young Neuro Scientist Programme of the Centre for Transdisciplinary Neurosciences Rostock (CTNR). T.K. received funding from the Deutsche Forschungsgemeinschaft (DFG, German Research Foundation, https://www.dfg.de/en) grant 225222086. The DFG or CTNR did not play a role in the study design, data collection and analysis, decision to publish, or preparation of the manuscript.

**Competing interests:** The authors have declared that no competing interests exist.

[2]. Intercellular communication in cell populations affects the intracellular behavior of individual cells. Signaling pathways include intracellular compartmental processes such as endocytosis or the transfection of lipoplexes. Although such processes can sometimes be approximated with static compartments, dynamic compartments are the natural method for precise and expressive models.

Several modeling approaches that support dynamic compartments have been proposed in the past [3–6]. However, in contrast to approaches limited to static compartments, running simulations for these modeling approaches proved too computationally challenging for usage in relevant biological applications. In fact, some of the approaches developed were never equipped with a (publicly available) simulator implementation.

In the following, we present an approach that overcomes these performance problems and makes dynamic compartments feasible for studying cellular processes. In particular, our contributions are:

- an analysis of rule-based language design choices on efficiently executing dynamic compartments,

- an analysis of execution requirements for an expressive rule-based language, such as ML-Rules, which allows compact and expressive specifications of models with dynamic compartments [7],

- a design of a new efficient stochastic simulation algorithm tailored to models with dynamic compartments combining various algorithms with code generation,

- an implementation of a new modeling and simulation tool ML-Rules 3 in Rust based on the new efficient stochastic simulation algorithm, and slight adaptations in comparison to earlier implementations of ML-Rules; the latter includes syntactic enhancements to improve usability, like type checking of units of measurements, and a new type of attributes to facilitate efficient simulation,

- a realization of a WebAssembly-based web simulation tool at http://mlrules.pages.dev to enable simple deployment of the tool,

- and an evaluation of the expressiveness and performance of ML-Rules 3 based on case studies from cell biology, including an mRNA delivery model, where based on dynamic compartments, due to ML-Rules 3, now closer matches to biological results become possible.

## 1.1 Compartmental dynamics

Cell biology considers compartments essential elements of cell behavior [8]. Compartments affect cellular processes by controlling both the reactions that can occur and the rate at which they do. They are considered important to modeling signaling pathways, in which extracellular ligands trigger a cascade of biochemical reactions that span various cellular compartments, such as membrane, cytosol, and nucleus [9]. A prominent example is the canonical Wnt/$\beta$-catenin signaling pathway, which is essential for cellular functions such as proliferation and differentiation and is involved in several diseases, including cancer [10]. In the last 20 years, more than 20 quantitative models have been built to analyze the Wnt/$\beta$-catenin signaling pathway [11]. Many of these models express the causal influence of compartments on processes, e.g., constrain processes to specific compartments or study proteins shuttling between compartments [12, 13]. Also, the volume of the compartment may influence the reactants' density and, consequently, the kinetics within the different compartments [9, 14]. Compartmental

(a)

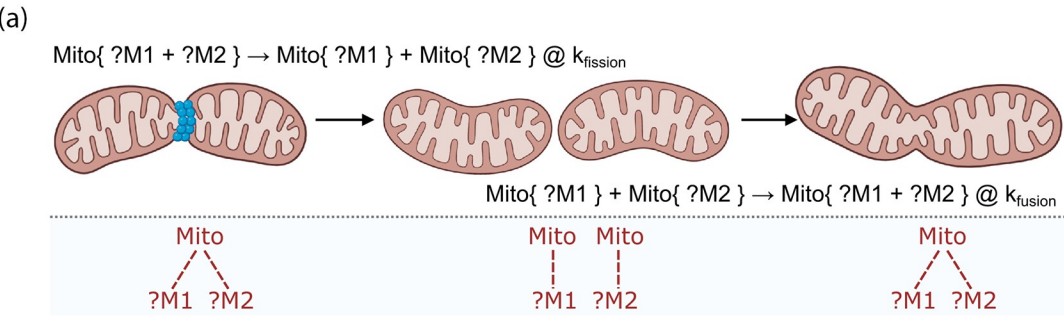

(b)

(c)

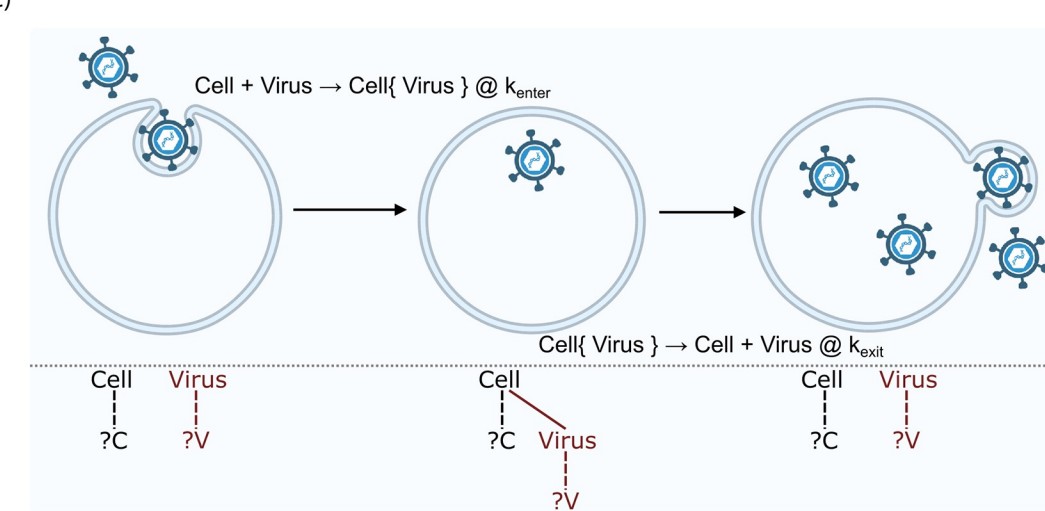

**Fig 1. Examples of compartmental dynamics in cell biology.** The figure shows a sketch of the processes, the specification in ML-Rules 3, and the n-ary tree structure of the term rewriting that underlies ML-Rules interpretation. The red lines in the n-ary tree structure indicate structures that are changed during the reaction. For the rules, we use the ML-Rules notation. The left side is transformed (using the →) into the right side at a rate (following the @ symbol). The {} denote nesting relationships. (**a**) Fission and fusion of compartments: A large mitochondrium can divide into two smaller ones, whereby its content (here ?M1

and ?M2) is distributed between the two new mitochondria. Two mitochondria can also fuse into a large one containing the solution of both smaller ones. (**b**) Creation and removal of compartments: During the endocytosis of the lipoplex, an endosome is formed around it. Inside the cell, the lipoplex can unpack its content (?L) into the cell. (**c**) Shuttling of compartments: A virus carrying DNA or RNA can enter a cell. New viruses produced in the cell can level it and spread.

constraints on reactions are supported by most simulation tools and standardized modeling exchange formats such as SBML [14].

Considering dynamic compartments opens up new possibilities. Treating compartments as regular species with attributes enables causal effects between compartments and of the compartmentalization on underlying processes, called *downward causation* [6]. Causality also occurs in the opposite direction— intra-compartmental dynamics can influence the compartmental level, including its attributes such as volume. This *upward causation* can also result in changes to the compartmental structure, such as fission and fusion, or the creation and deletion of compartments.

The changes in the compartmental hierarchy occur during the simulation execution. Therefore, they have been called *dynamic compartments* [15]. Dynamic compartments are one form of variable structure model [16] and might occur at and involve various levels of cellular organization [6]. Signaling interactions within multi-cellular populations synchronize and organize the individual cells' behavior in dynamic processes such as tumor growth [17] or morphogenesis [18]. Cell proliferation, migration, and differentiation are regulated by several signaling pathways, which again react to changes in the cells' environment. Dynamic compartments are also necessary for intracellular dynamics; during the last decades, it became evident that a static view of intracellular compartments does not suffice [8]. The internalization of receptors is crucial in regulating signaling pathways [19]. Receptors are sorted into different endosomal compartments when entering the endocytic pathway. To accomplish this sorting, the organelles undergo frequent compartmental dynamics such as maturation, transformation, fusion, fission, and degradation [20]. More recently, the development of mRNA vaccinations has increased the interest in studying liposomal dynamics [21] and its modeling [22] to provide effective drug delivery systems.

Compartmental dynamics take on various forms, as also discussed in [23] and modeling approaches such as Bioambients [3], Brane Calculi [24], or BetaBinders [25], including a) the fission and fusion processes of compartments, as in the case of mitochondria [26] (Fig 1a), b) the formation and disintegration of cellular compartments, as in the case of liposomal dynamics [27] (Fig 1b), and c) a compartment entering or leaving another compartment, as in the case of a viral entry or release [28] (Fig 1c).

## 1.2 Stochastic simulation

To simulate these compartmental dynamics, we will interpret compartments as discrete entities. We assume that the discrete compartments can be arbitrarily nested, that entities can exist within and outside of compartments, that compartmental dynamics rely on the explicit definition of reactions, and that the reactants are well-mixed within and outside of each compartment.

Based on these assumptions, adopting stochastic simulation algorithms (SSA), popularized under the term of Gillespie algorithms [29], appears most appropriate. These algorithms can be formalized using Continuous-Time Markov Chains (CTMCs) [30]. In CTMCs, the distributions for the waiting time until the next state transition and the successor state only depend on the current state. The sojourn time in a state is exponentially distributed, similar to a

physical decay process. The propensity is the expected rate of firing for a particular transition, given a specific model state. Each potential transition $i$ between states in the CTMC has a propensity $p_i$.

In the *direct method* family of methods [31], a timestep is sampled using the exponential distribution. The total propensity sum $\Sigma_i p_i$ is used as the rate parameter of the exponential distribution. The reaction is selected by weighted random choice ($\mathcal{P}(i)$ $p_i$). The *first reaction* family of methods [29] computes a time for every possible reaction by sampling from the exponential distribution (with the respective $p_i$s) and then selecting the smallest one. Both original approaches are wasteful, as many computations might be needlessly redone. Major performance benefits result from storing results, such as propensities [32] or the time of the next event of a reaction [33], and updating this information on demand. For this update on demand, a dependency graph stores the dependencies between reactions and the associated propensities. The next reaction method [33] builds on the first reaction method and maintains a schedule of reactions and the time they will occur. After a reaction is executed, based on the dependency graph, the affected reactions are rescheduled. Another approach builds on the direct method, stores the propensities of reactions, and, based on the dependency graph, updates the affected propensities in each step [32].

## 1.3 Rule-based modeling and compartments

The modeler enumerates all possible species and reactions in species reaction models. This modeling approach is limited when dealing with complex systems, like proteins involving multiple binding sites, as the number of possible species and reactions can grow exponentially in these cases [34]. The fundamental idea of rule-based modeling (combined with CTMC semantics) is to use rewriting rules to specify transition classes of a CTMC. The left side of a rewriting rule specifies the reactants and their context as a pattern matched to the current state. Using patterns, a single rule can express a large set of reactions, which can be parametrized with the variables matched in the pattern. The patterns on the left rule side constrain the reactants participating in a reaction, for example, a pattern like $A(x) + B(x) \rightarrow \dots$ expresses that one entity $A$ and one entity $B$ can only react if they share an attribute value $x$. Each successful match contributes one transition within the CTMC. The rewriting rule is annotated with a rate expression, which is evaluated based on the left side's match. This can include factoring mass action kinetics into the transition rate (based on the multiplicities of the reactants) as well as employing attribute values of the reactants or their context. The successor state of the resulting transition is computed by applying the rewriting operation: removing the reactants and adding the products.

Including dynamic compartments poses a challenge to rule-based modeling languages. Languages like Kappa [35, 36] or BNGL (as part of BioNetGen) [37, 38], for example, are based on graph rewriting rules. Their focus is on modeling interacting entities between which bonds are created and destroyed. Bonds are defined for exactly two entities, which precludes using them to express compartments. Static compartments can be integrated [9, 39]. They are used to restrict the scope within which a reaction takes place. Shuttling of simple (or bound) molecules between static compartments as they are commonly supported also by the modeling standards [14], can be easily expressed, e.g., by *Protein@Cytosol → Protein@Nucleus* in BioNetGen. However, here, neither the cytosol nor the nucleus forms a species and, as such, a potential reactant or product of a rule or reaction.

Dynamic compartments can be expressed using graph rewriting when extending the formalism with hyperedges. Hyperedges are those that can connect to more than two vertices. This approach is used in the modeling formalism React(C) [4]. Here, a compartment can be

denoted by a hyperedge, which can join any number of entities. In addition, the compartment itself can be represented as a species. The following rule describes a cell containing a nucleus and a protein that shuttles into the nucleus (Fig 1c):

$$Cell(c) + Nucleus(n, c) + Protein(c) \rightarrow Cell(c) + Nucleus(n, c) + Protein(n) \ .$$

Note that the containment relation is expressed purely via variables $c$ and $n$, identifying the cell and nucleus as a compartment (hyperedge). So the values of the variables $c$ and $n$ are identifiers for the respective compartments and as such unique. Entities not affected by the rule can be omitted on both rule sides, Therefore, we could have omitted $Cell(c)$. However, this would have made it harder to understand the role of the value of variable $c$.

In the approaches based on hypergraphs, new compartments can be created by introducing new, unused ("fresh") values for variables on the right side. Similar solutions to model dynamic compartments can be found in process algebras, e.g., the attributed PI calculus [40]. The following rule expresses the fission of an organelle, with the $\nu$ operator yielding fresh values (identifiers) for the two new cells, i.e., $c_1$ and $c_2$:

$$Cell(c) \rightarrow (\nu c_1, c_2)Cell(c_1) + Cell(c_2)$$

However, this cell fission example also shows a fundamental problem with the hypergraph approach: the effects on the contained entities can not be modeled as easily as the effect on the compartments. In the above reaction, all entities referencing the value of $c$ as their containing compartment need to be updated to either $c_1$ or $c_2$. Also the fusion of compartments

$$Cell(c_1) + Cell(c_2) \rightarrow (\nu c)Cell(c)$$

would require that all entities referencing the values $c_1$ and $c_2$ would now need to reference $c$. These changes can be expressed with workarounds, for example, by defining intermediate states and additional rules for each species contained within the original compartment, with infinite propensities, but this clutters the model description and is computationally rather expensive. An additional problem with the extension of graph rewriting to hypergraphs is that finding occurrences of the left rule side in the current state is harder. Whereas graph rewriting can exploit some properties (e.g., Kappa exploits "rigidity" [41]) to speed up this process, similar optimizations for hypergraphs are not known. As a consequence, no simulation tool based on hypergraph rewriting has been published. In particular, no simulator for React(C) is available.

An alternative approach to graph rewriting is multiset rewriting, a special case of term rewriting [42, 43]. Here, the containment in compartments is structurally equivalient to an $n$-ary trees (see also Fig 1). Therefore, they can be expressed as terms with a variadic function symbol or, equivalently, an associative and commutative function symbol for forming multisets [44, 45]. The contents of a compartmental entity can then be represented by a multiset, usually denoted with the symbol + or, in infix notation. The protein movement rule above (Fig 1c) can be written as

$$Cell(Nucleus(?n) + Protein + ?c) \rightarrow Cell(Nucleus(Protein + ?n) + ?c) \ ,$$

where $?c$ captures the remaining multiset within the cell after finding a nucleus and a protein in a cell, and $?n$ captures the contents of the nucleus. These variables starting with ? (called "sequence variables" in the rewriting literature [46]) play a central role, as they denote the entities unaffected by a rule and allow operations on those multisets of entities, i.e., the content of the compartments [47]. One line of work that very closely follows the idea of multiset rewriting

is Colored Stochastic Multilevel Multiset Rewriting (CSMMR) [5, 48]; another is ML-Rules [6, 49].

In multi-set rewriting, the cell fission rule could be written as

$$C(?c) \rightarrow C(?c_1) + C(?c_2) \textbf{ where } (?c_1, ?c_2) = splitHalf(?c) \ ,$$

binding the result of the function *splitHalf* to a pair of variables in a **where** block [47]. In the syntax of CSMMR, the rule would read as

$$C(x) \rightarrow let(z, y) = x \ in \ C(z) + C(y) \ .$$

In the syntax of ML-Rules 3, the operator "+" is overloaded to realize an inline function on cell content, which randomly assigns each element of $C$ to bind either to the sequence variable $?c_1$ or $?c_2$, i.e.,

$$C(?c_1 + ?c_2) \rightarrow C(?c_1) + C(?c_2)$$

(see Fig 1a). In some biological processes, fission events occur asymmetrically. For example, such asymmetrical fission events are observed in mitochondria. The small mitochondria might include many damaged parts and can be removed from the cell, thereby contributing to the health of the mitochondrial network. To model this process weights can be assigned to the sequence variables

$$Mito(?m_1[R_1] + ?m_2[R_2]) \rightarrow Mito(?m_1) + Mito(?m_2)$$

where $R_1/(R_1 + R_2)$ and $R_2/(R_1 + R_2)$ are the ratios of elements assigned to $?m_1$ and $?m_2$, respectively.

While multi-set rewriting provides an expressive and readable way to model dynamic compartments (see Fig 1), languages that are built on multi-set rewriting, such as ML-Rules, are still hard to execute efficiently precisely because of their expressive power. One approach to alleviate this is to provide specialized simulators that exploit that some language features are not used in a given model [49], thus working on a subclass of models. Another subclass is considered in the modeling approach presented in [15]: it only considers models where compartments are not nested, and entities do not exist outside compartments.

## 2 Simulation engine

We implemented version 3 of ML-Rules using the Rust programming language to develop and test an efficient simulator for dynamic compartments. The model is specified within an external domain-specific language which is parsed into Rust code. The structure of the implementation and its components is shown in Fig 2. We have tuned the implementation for performance on the main expected code paths. That means we specifically applied several optimizations to the typical main loop at the potential cost of the unusual behavior. Overall optimizations include minimizing allocations, a flat, indexed-based data layout, and partial evaluation of repeated computations. The following sections will discuss differences from the previous versions and their implications on runtime performance and user experience.

Compared to the previous implementation, we introduced a few changes to the syntax of ML-Rules (Section 2.1). However, the focus of our research has been to develop a simulation approach that is able to handle models that may exhibit a wide range of compartmental dynamics efficiently. It incorporates a new variant of an SSA specifically designed for large SSA systems (Section 2.3). This variant of the SSA (labeled flat model simulator in Fig 2) is not specific to compartmental models, as compartmentalization is abstracted away in a previous step. One significant optimization is code generation for repeated expression evaluation 2.4.

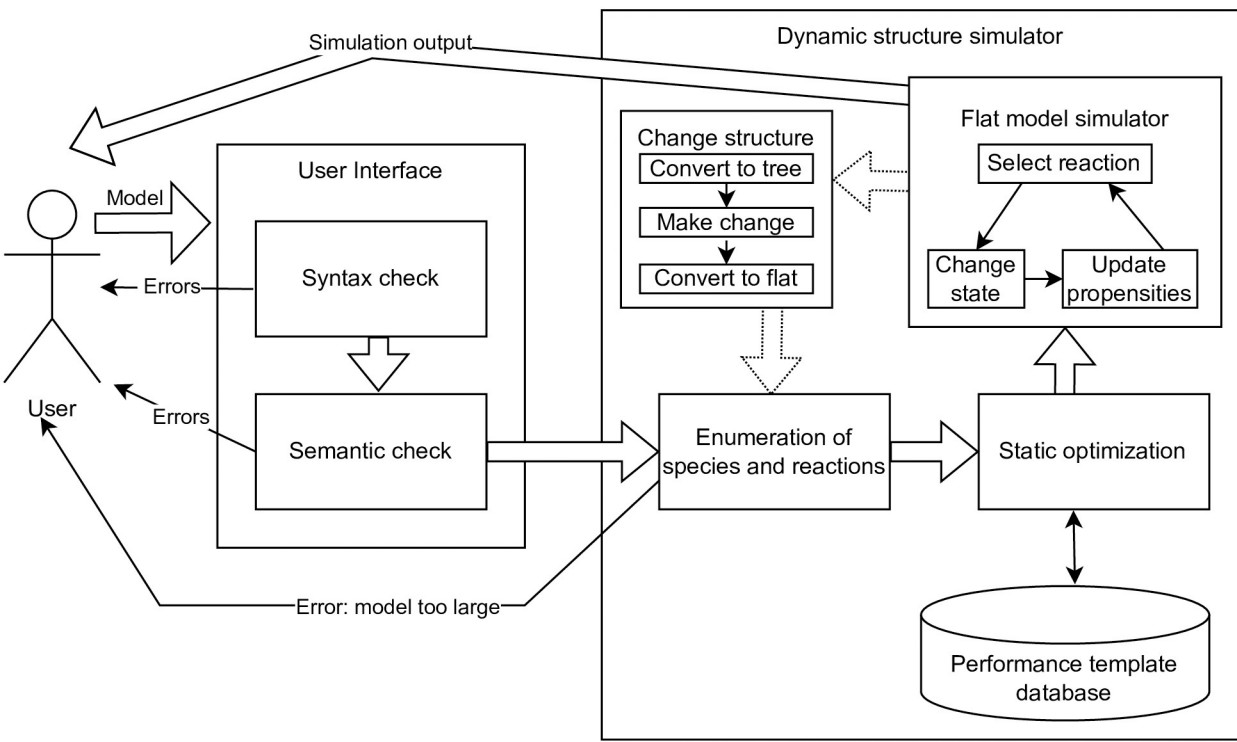

**Fig 2. This Figure shows the flow of information between components when running a simulation.** The user specifies the model, which is checked for syntactic and semantic consistency (like units or variable names). This is done through interaction with the web editor or command line interface. If these checks fail, errors are returned. Otherwise, the simulation loop starts by enumerating all possible species and reactions within the system into a flat representation. This can lead to errors if the system is too large. This flat representation is then optimized and put into the flat model (non-compartmental) simulator. Whenever a dynamic structural change is needed, the flattened model is transformed into a hierarchical, compartmental representation to execute those changes, and the loop starts again with the enumeration.

We also incorporated a hybrid rule-based simulation method, allowing individual attributes to be simulated in a network-free manner (Section 2.1). Finally, through the use of recent advances in web technology (namely WebAssembly), we can provide a simple web editor that enables fast local execution using the same codebase (Section 2.5).

## 2.1 Language

ML-Rules 3 builds on and adapts the language ML-Rules for rule-based multi-level modeling of cell biological systems [6] and its formal semantics [47]. The model is comprised of

- constants that can be used as input parameters for simulation experiments,

- definitions of functions for simplification of repeated notations,

- the initial model state,

- rewriting rules, that can also change the structure of compartments and may use complex expressions for the rates and attribute values, and

- a definition of potential outputs of the model.

The rewriting rules have the form:

$$\texttt{<left> -> <right> @ <rate>}$$

We have a left side transformed (using the arrow) into the right side at a rate (following the @ symbol). The {} denote nesting relationships.

In the following, we describe some key aspects of ML-Rules and some slight changes compared to earlier implementations. The syntactic enhancements (with the exception of the network-free attributes) were not intended to improve the simulation performance but modeling in ML-Rules. A core aspect of the language design is the nature of the patterns on the rules' left side. Modeling languages based on graph rewriting, such as Kappa or BNGL, employ graph patterns. Modeling languages based on multiset rewriting, such as CSMMR or ML-Rules, employ term patterns whose matching can be considered a specific case of unification [50]. The matching relates to compartmental structures and attributes [6, 49]. Attributes can be addressed either by position (structural pattern matching) or by name.

For a species

$$S(att\_1: int, att\_2: String)$$

the previous ML-Rules versions required to list all attributes of a species:

$$S(a1,a2) -> S(a1+1, a2)$$

as attributes were identified by their position. Instead, we now have named attributes, where, in the case above, only the changed attribute needs to be listed:

$$S() -> S(a1=S.a1+1)$$

From a formal perspective, rule-based approaches implicitly match omitted attributes. For example, given a cell species that has two attributes denoting its phase and volume, the rule

$$Cell(phase == 1) -> Cell(phase=2)$$

would be implicitly extended to

$$Cell(phase == 1, volume == v) -> Cell(phase=2, volume=v)$$

to express that the volume does not change. Named attributes have been used in other tools like BioNetGen [38] or Chromar [51].

The use of named attributes is obviously particularly useful if species have many attributes. For example, a simulation model of the Baltic Cod, which was developed in a previous version of ML-Rules, relied on structural pattern matching [52], and is available at http://github.com/Baltic-Cod/EBC_IBM/blob/main/Basic_asph/Basic_asphyx.mlrj, has been rewritten using ML-rules 3 to exploit the more efficient execution. This also resulted in a more succinct representation due to the named attributes http://mlrules.pages.dev/gm/4/day.

Typically, the simulation engine for rule-based models transforms rules into reaction networks by enumerating all possible values of attributes in advance to speed up the actual simulation [53]. Attributes that may assume many different and potentially unbound possible values are a challenge. In the case of unbound continuous attribute values, e.g., if the size of a lipid raft changes depending on the number of membrane proteins being aggregated within the raft or due to merging [54], this *in-advance-enumeration* becomes impossible. Even finite, categorical values can lead to a combinatoric explosion in the number of reactions and thus increase the reaction network size beyond tractable limits. This results in a *model too large* error in Fig 2. We have introduced specific types for this kind of attribute called network-free-integer and network-free-continuous. If an attribute is defined as one of these types, its values will not be explicitly enumerated before simulation execution (to generate the reaction network). For simulation, the attribute of type network free is equipped with a vector that stores the currently

existing attribute values when a reaction fires. Once a rule fires, the species is instantiated with appropriate values. The approach is similar to the network-free simulation in BioNetGen [55] or CSMMR [5], particularly to the hybrid approach introduced in [56] that combines network-free and network-based calculations. However, instead of specifying entire species as network-free species, the modeler declares individual attributes to be network-free, and the simulator handles only those attributes or combinations as network-free. This distinction between being executed as network-based or network-free applies only to attributes of species that do not form a compartment. All compartments are handled as individual entities, as compartments are typically characterized by an attribute of continuous type, i.e., the volume, and they may contain an arbitrary number of diverse species (which again might be attributed), so one compartment's state is very likely different from the next and is (including its attributes' values) treated individually.

Other adaptations compared to earlier ML-Rule implementations are motivated by further increasing readability. In the previous ML-Rules versions, only equality constraints on the attribute values could be expressed on the left side of the rule. For example,

$$\texttt{Cell(vol, M)} -> \ldots \texttt{if vol} > 100 \texttt{ than k1 else 0}$$

can be expressed in ML-Rules 3 as

$$\texttt{Cell(vol} > 100, \texttt{phase} == \texttt{M)} -> \ldots \texttt{@ k1}$$

Further adaptations aim to provide further support to define correct models. Similarly, as in other tools [57], modelers are now asked to assign units of measurement to numerical variables and constants, like micrometer, 1/second, and liter. The simulator performs the proper conversion and automatic type checking [58] (Fig 2). This means a meter is correctly added to centimeters but not seconds. In addition, we introduced an enum type, allowing the modeler to constrain the admissible string values for a specific attribute to a specific set. Type inference, as well as constraint checks, occur ahead of simulation execution.

## 2.2 Efficient handling of dynamic structure reactions

As discussed in Sections 1.1 and 1.3, dynamic changes in the system's structure are core to the ML-Rules modeling language. Their efficient execution has been a challenge in the past. Most simulation tools assume static compartmental structures, and their simulators have been optimized accordingly [36, 38, 57]. Integrating compartmental dynamics into these simulators without significant loss of efficiency has been identified as a daunting challenge [59].

One option to solve the problem of dynamic structural changes is to write a fully dynamic simulator. This simulator traverses the tree structure of the model after each reaction execution to instantiate new reactions and re-calculate propensities on demand. It also checks whether the dependency graph requires any updates. This is how a previous implementation of ML-Rules was realized [49] and what is shown in the left panel of Fig 3. However, compared to methods optimized for static networks [53], this has a significant overhead.

For ML-Rules 3, we implemented a new hybrid simulator approach shown on the right panel of Fig 3. During an additional analysis step, every reaction is analyzed and marked if its execution would result in a structural change. In the models we encountered, most reactions do not induce a structural change. These reactions are called regular reactions. A similar distinction between regular and complex reactions has been made when simulating part of the system using deterministic numerical integration methods [60] or by tau-leaping [61].

The primary mode of our simulator is running an SSA (Section 2.3) that executes only regular (non-dynamic) reactions. The model representation has been flattened, enabling static

Network-free algorithm ML-Rules 3 algorithm

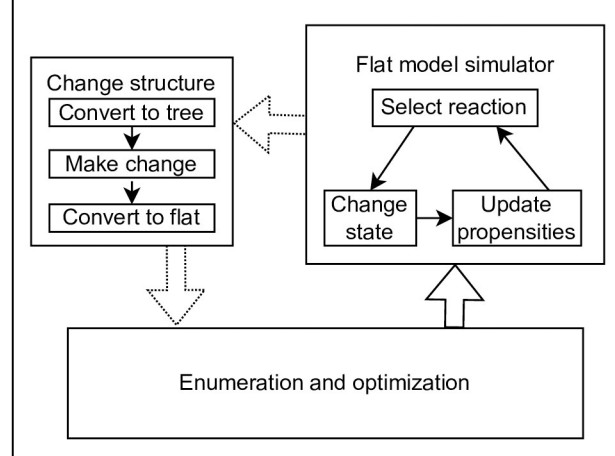

**Fig 3. Comparison of the dynamic structure approaches.** On the left, we have the previous network-free method. It is simpler to implement, but due to the dynamic structure, each step is significantly more costly in terms of performance. For example, neighborhood relations are dynamic, and therefore, the nesting tree structure needs to be traversed. Our approach (a subset of Fig 2) is on the right. We can utilize a faster flat (i.e., static) model simulator and have a costly but amortizing transformation and optimization process on the rare occasion of a structural change. For example, all relations can be encoded by static indices.

index-based access to the model's state, i.e., every species' amount is stored in an array. Hierarchical relations are only preserved to the point where they are needed to reverse the system representation to a tree form. But they are neither accessed nor needed during the simulation as long as no transformation to a tree form is required. Therefore, we call this the *Flat model simulator* in Figs 2 and 3. When we encounter a structure-changing reaction, the model is transformed back into the tree representation reflecting the compartmental structures. The reaction is applied to this structural representation of the model. The resulting structured state is then transformed back into a flat representation, and processing is continued. This circular process is shown on the right side of Fig 3. This Figure shows the inner loop within the flat model simulator and the loop involving the dynamic structure changes. Every dynamic structure change requires an entire rebuild of the simulator, including re-enumerating all potential reactions and a do-over of the static optimization phase. Especially for larger systems, this can be relatively costly; however, as long as dynamics structural changes are rare, the costs amortize. We will analyze this in the evaluation section 3.

## 2.3 Stochastic simulation algorithm for static compartments

Based on our experience with different SSAs, we developed our variant, which is suited particularly well for typical problems in ML-Rules, where systems can be very large, e.g., for simulating multi-cellular models. Systems become particularly large when identical reactions are replicated across many different compartments. This is the *Flat model simulator* in Fig 2. The implementation is based on storing the propensities in a binary tree with cumulative sum tracking [62]. Each node in the tree corresponds to one potential reaction and stores two values: The propensity of that reaction and the sum of all propensities of its child nodes. This allows for updates of the total propensity sum with logarithmic complexity changing a single propensity. Furthermore, reaction selection (a weighted random choice based on the reaction propensities) can also be completed in logarithmic time complexity. Similar approaches are

used in other algorithms like the logarithmic direct method [63] or the simulator implementation for $S\pi@$ [64].

An additional advantage of the tree-based approach is its improved numerical stability due to the fewer operations needed to update the propensities. Another way to minimize total propensity calculation time is by keeping track of the total propensity sum and only adding and subtracting the changes to reaction propensities after reaction execution as done by the Optimized Direct Method [32]. However, here, numerical errors from floating-point addition and subtraction accumulate over time and need to be dealt with. In our tree-based approach, numerical rounding errors only originate from the single summation, similar (arguably even better) to a single linear summation approach. The numerical error does not accumulate over time. It is independent of the number of steps and depends only on the number of reactions.

After a reaction has fired, multiple propensities in the tree must be updated. In principle, each update in the tree could be done individually based on the dependency graph via updating the cumulative sum until we reach the root. Instead, we roll these changes out in two phases. First, the node values are changed based on the dependency graph. Afterwards, the cumulative sums are updated as needed. If one were to update all sums individually, some nodes near the root might perform repeated summation updates. What total summation operations need to be done for each reaction is calculated ahead of time in the static optimization component (Fig 2). This computation has been accelerated via the use of bitsets. The generation of the dependency graph and the propensity tree must be efficient, as it needs to be redone after every non-regular reaction. Reaction selection is based on a weighted random choice based on the reaction propensities stored in the tree. The performance of reaction selection is improved by sorting the more likely reactions towards the root of the tree.

## 2.4 Performance templates

ML-Rules is an external domain-specific language. This means that all expressions are written in a custom text format. This format is parsed by the simulation tool. One of the main downsides of using an external domain-specific language is the significant computational overhead in the repeated evaluation of mathematical expressions. Generally, the more expressive and generic a domain-specific language is, the harder it is to execute efficiently. In a more constrained language, code and algorithmic variations that are tailored to the specific requirements of the application and hardware architecture are easier to achieve. However, a domain-specific language is typically more useful if it is more generic and expressive.

In [49], we developed specialized simulators for specific sub-classes of ML-Rules models, e.g., those that do not exploit compartmental dynamics. In [65], we developed an approach generating an entire simulator optimized for a specific model defined in a rule-based language, such as BioNetGen [37]. After parsing the model, custom C or Rust code was generated. This code was then compiled and optimized using existing compiler software, resulting in a high throughput performance. This technique (called *partial evaluation* or *Futamura projection*) [66, 67] is an established technique to deal with the problem that genericity does not go along well with achieving performance.

The central idea of partial evaluation is that a function with multiple inputs can be reduced to a simpler version if some of the inputs are known. For example, the power function $f(base, exponent)$ may be simplified to $f(base, 2) = base \cdot base$ for the case of a known exponent. The two core applications of this in the context of SSA are the dependency graph and the propensity calculations. For the dependency graph, data structures and loops can be omitted. With the propensity calculations, optimizations focus on reducing mathematical expressions and removing overhead from dynamic interpretation.

Adopting the approach developed in [65] for dynamic compartments appears impractical. Its performance gain relied to a large degree on optimizing the updates of the dependency network. With dynamic compartments, these reaction networks change during execution; thus, optimization (recompilation) would need to be repeated after each structural change. However, the (re-)compilation induces a significant overhead. Especially for large models (which is typical for ML-Rules models), the additional compilation steps on every model run increase runtime significantly, independently of model execution duration. Therefore, we developed an approach that combines dynamic interpretation and partial evaluation and does not require repeated compilation on every change. Our approach focuses on the rate evaluation (and thus propensity updates). This is costly for most dynamic compartmental models, as rate expressions involving dynamic compartments are typically complex. The complex expressions result in large abstract syntax trees (AST) that need to be parsed during execution. When using an interpreted or reflection-capable language like Java, code generation for these expressions can be introduced relatively simply [68]. For compiled languages such as Rust, a different approach is required. Our approach developed for ML-Rules 3 generates performance templates during the simulation and stores them for later reuse. Every time the simulator encounters a new AST, it checks whether generated code in the form of a performance template is available to replace this tree. Each performance template presents a partially evaluated AST and can be parameterized with numerical values (like model-specific constants) and indices (as used to identify involved species). If such an optimized previous version is found, it is used instead of the AST. If not, the simulator dynamically interprets the expression (by evaluating the AST). It generates an optimized code for future use and stores the corresponding performance template for later reuse. With every new model developed or recompilation of the model, the simulator can reuse previously created templates.

## 2.5 Web editor

The software tool is primarily designed as a command line tool. However, as a proof of concept, we also built a web-based version. The idea to run simulations on the web is not new and has been around almost as long as the web itself [69]. There have also been attempts at running stochastic simulations of biochemical models using a web interface [70].

However, the execution has been typically delegated to a server in the background or the cloud to enable an efficient simulation of models [71]. WebAssembly is a binary format recently developed that enables fast execution in web browsers [72]. WebAssembly is executed on a stack-based virtual machine similar to other native code. Additionally, it interoperates well with JavaScript. WebAssembly allows reasonable performance for simulation without installation in a web browser or major changes to the underlying simulator code [73].

As the Rust language can be compiled into WebAssembly, we can use the same code base for the simulator and only need to add a small frontend. We used the Rust Yew framework (http://yew.rs/) for this. A two-panel user interface is provided at http://mlrules.pages.dev, which includes a code editor with some syntax highlighting based on the Monaco editor from Visual Studio Code and another panel to display simulation results or any errors and instructions from the simulator.

A significant advantage of this WebAssembly approach is that there are next to no host costs. The simulation is executed on the end user's machine. The web page is static and has a size of roughly 10 Megabytes, which is very cheap or free to host compared to other approaches that run the simulation on the server. We also added the possibility to link to specific models. We use this in our case studies to provide links to the various models, which facilitates the reproduction of results and allows users to change and adapt models and conduct

basic experiments with no additional setup. As done throughout this paper, linking to an executable model with a custom URL is possible.

## 3 Evaluation

In this section, cell biological simulation models showcase the capabilities and performance of ML-Rules 3. All performance experiments were conducted using an intel i9-13900K CPU, running Rust 1.74. WebAssembly was run in Firefox version 120 using its spidermonkey engine.

### 3.1 Fission yeast model

The first case study is based on a multi-cellular model of fission yeast, including cell division and mating type switching depending on intracellular dynamics. The model is available at http://mlrules.pages.dev/yeast/300/min. The model (see Fig 4) has been adopted from the original ML-Rules publication [6] to show the expressiveness of ML-Rules. The fission yeast model includes an early cell cycle model [2].

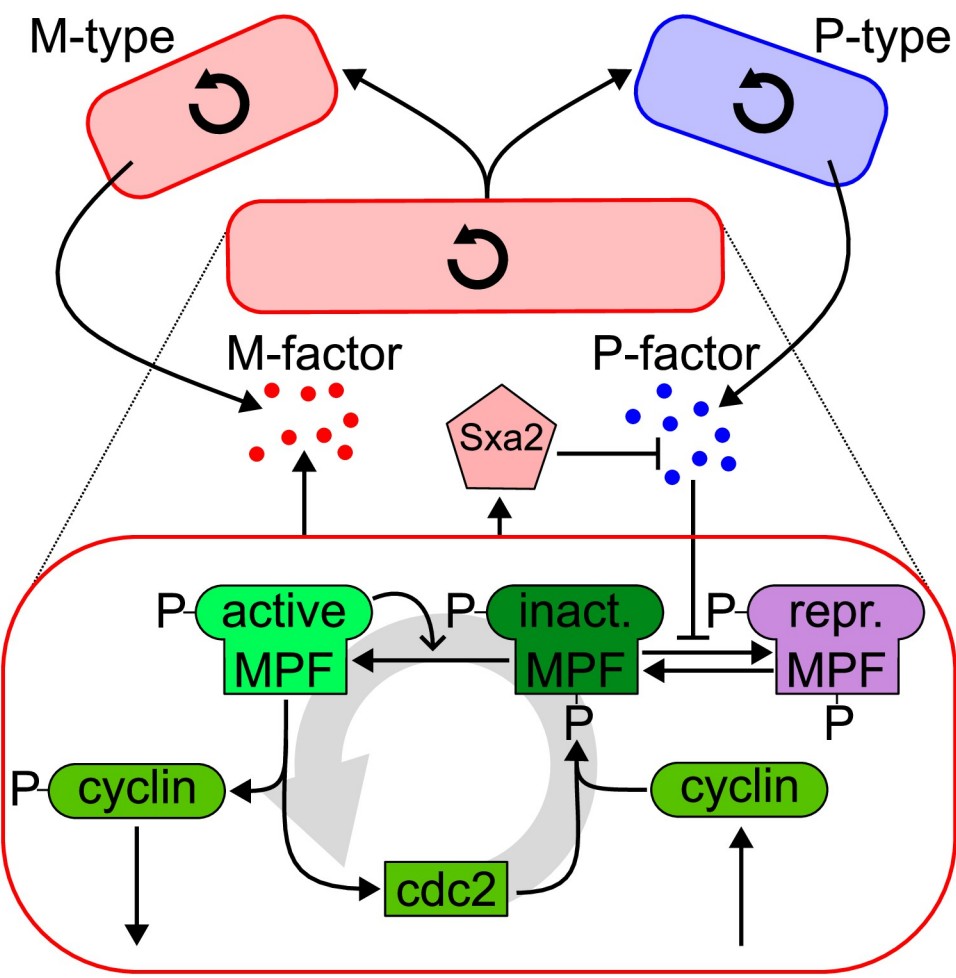

**Fig 4. Fission yeast model.** Inside the cell, proteins oscillate. Outside the cell, Sxa2 and pheromones (P and M-type) can inhibit the cell cycle of cells of the opposite mating type. Triggered by the cell cycle, a yeast cell can fission into two daughter cells.

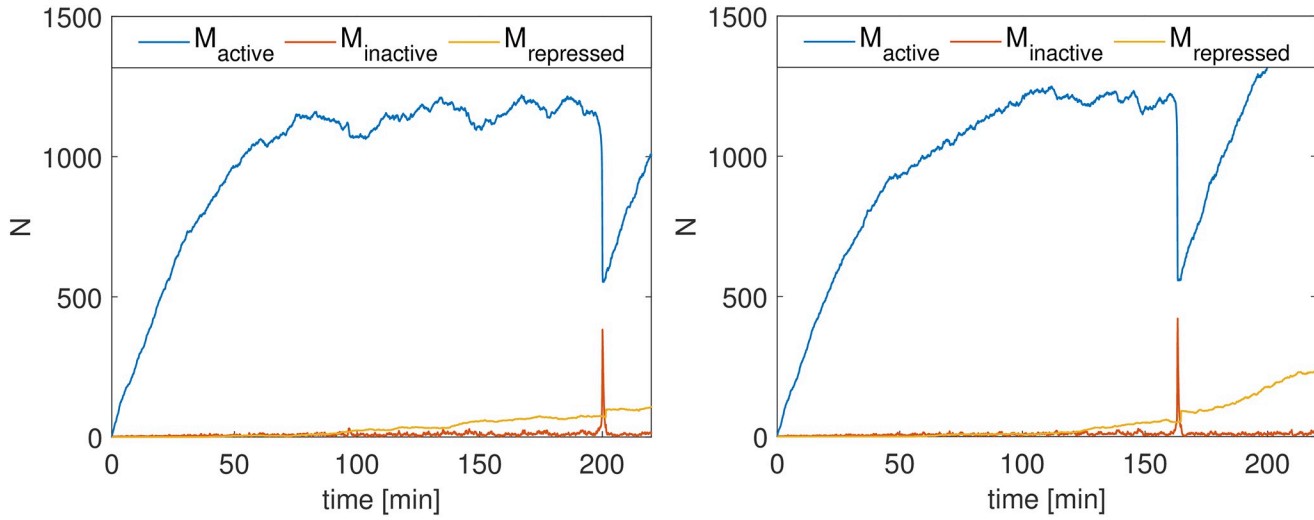

**Fig 5. Two sample replications of the fission yeast model show the oscillation of active, inactive, and restricted MPF.** At 200 minutes and 163 minutes, respectively, a fission event is triggered by the spike of inactive MPF.

The cell cycle model consists of two proteins (cyclin and cdc2) that can form the maturation-promoting factor (MPF). MPF exists in three versions (active, inactive, and repressed), which oscillate with a cycle duration of approximately 200 minutes (see Fig 5). This oscillation triggers the cell to change between its phases (G1, SG2, and M), and eventually, a spike in inactive MPF causes cells in its M-phase to divide into two daughter cells. In addition, cells are characterized by a mating type (P or M) that might differ in one daughter cell from the type of the mother cell. Based on the mating type, cells produce and secrete pheromones (M- or P-factor) to inhibit the cell cycle of the opposite mating type cells by changing MPF from its inactive to the repressed variant. M-type cells also release P-factor-specific protease (Sxa2) that inhibits the P-factor pheromone.

This model relies on a unique combination of ML-Rules features and its expressiveness. Individual cells are defined as compartments that frequently divide. Compartments and proteins are equipped with attributes, e.g., to denote the cell cycle phase, the cyclin's phosphorylation state, or the cell's volume. The cells secrete pheromones into the extracellular environment. They influence the cell cycle of other cells of the opposite mating type. The kinetics depend on rate factors that require complex expressions, e.g., a Hill-type sigmoidal response curve defines the MPF repression (depending on the number of pheromones).

We have used this model as a performance benchmark. The reaction throughput rates for a run until 1000 minutes (simulation time) are shown in Table 1. ML-Rules 3 is significantly

**Table 1. Run time and reactions per second for different simulators.** The fission yeast model is executed until 1000 minutes (20 replications). The values in parenthesis refer to executing ML-Rules 3 as WebAssembly code. The ML-Rules 2 implementation of the model contains ten species and 20 rules. The ML-Rules 3 version of the model consists of 7 species and 20 rules. The model starts with two cell compartments and ends with about 12. During the simulation time, the model undergoes about 10 structural changes and half a million static reactions.

| simulator | runtime [s] | throughput [1000/s] |
|---|---|---|
| ML-Rules 2 | 12.4 | 40.8 |
| ML-Rules 3 | 0.124 (0.190) | 4190 (2690) |
| ML-Rules 3—network free | 0.321 (0.352) | 1590 (1440) |
| ML-Rules 3—w/o templates | 0.259 (0.331) | 1950 (1530) |

faster than the previous Java implementation, ML-Rules 2. If performance templates are used, we observe a roughly 2-order speedup. Even without the templates, the performance is 26 to 18 times faster. The 18x speedup is observed when enabling network-free attributes (see Section 2.1). The model does not use network-free attributes; even only enabling this capability introduces more branching in the critical code and slows down execution. We also tested the WebAssembly version. The data for this plot can be generated locally by visiting http://mlrules.pages.dev/benchmark/yeast/3/1000. The first number is the number of replications, and the second is the longest test duration in minutes of simulation time. The WebAssembly has a performance penalty of only about 10% to 50%.

## 3.2 mRNA delivery model

Our second case study is centered around the delivery of mRNA into cells. Understanding how to deliver mRNA into cells is crucial for its use as a drug or vaccine [21]. Ligon et al. [74] published a simulation model capable of simulating the mRNA delivery based on lipoplexes (small lipid spheres containing mRNA).

In their model, shown in Fig 6, lipoplexes are present within the extra-cellular environment for an hour before being removed by an event mimicking the cell's washing. During this time, clathrin-coated pits containing one or more lipoplexes form at the cell's surface. Via the invagination of the plasma membrane, endosomes are formed, and the lipoplexes enter the cell. Inside the cell, the endosome releases the lipoplexes, which then unpack their mRNA. Afterward, the mRNA can be translated into proteins.

With their model, the authors could gain insight into mRNA delivery by lipoplexes and the dose-response relationship. However, they also state a few simplifications and workarounds needed in their model, as the used modeling and simulation method only supported static compartments. One simplification is that all lipoplexes carry the same amount of mRNA in the model. In wet lab experiments, this number has been found to vary due to the different sizes of the lipoplexes [74, 75].

One possibility to model lipoplexes more realistically is to model each lipoplex as a compartment that contains the mRNA and enters the cell (which is also represented as a compartment). For this, support of dynamic compartments is required. Another possibility is to model

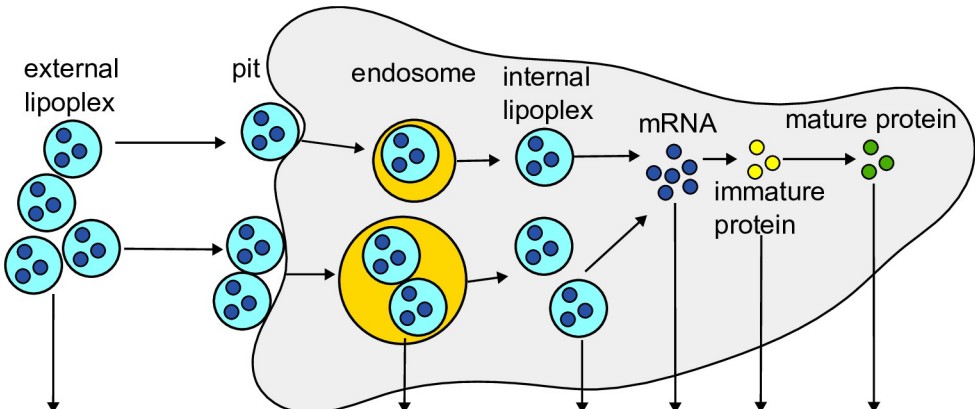

**Fig 6. Sketch of the mRNA delivery model.** Lipoplexes are present in the cell's environment and can accumulate in pits in the cell membrane. The lipoplexes in the pit can enter the cell via endocytosis, and the endosome can lyse to release the lipoplexes into the cell. Once they unpack their mRNA, the mRNA gets translated to proteins. The arrows that point down indicate the degradation of the species.

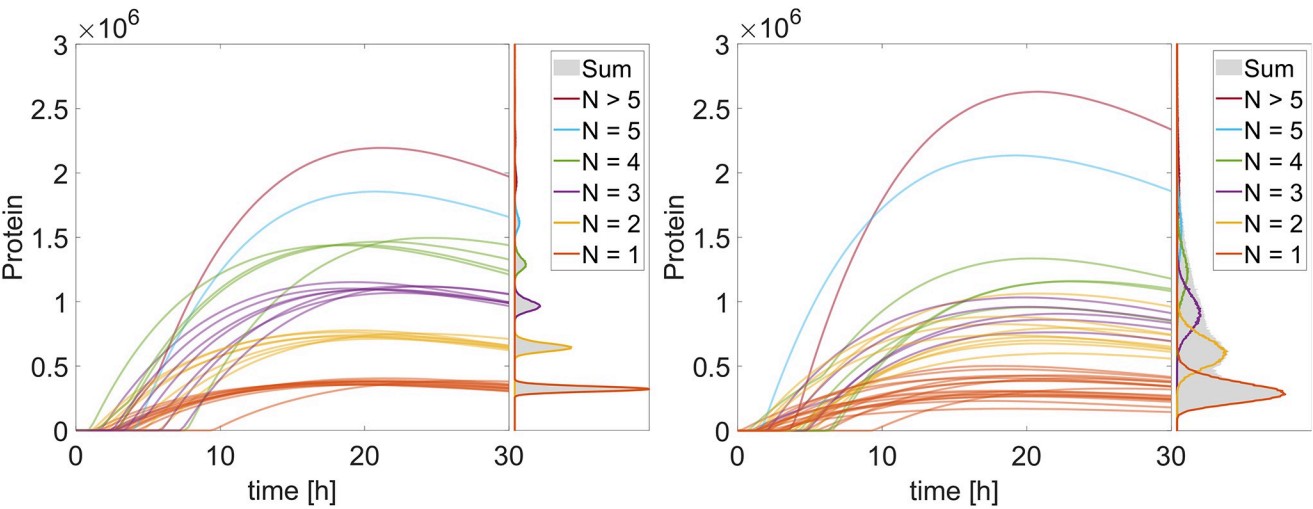

**Fig 7. Protein amounts over time for the original (left) and modified (right) mRNA delivery model.** The time courses are for 100 replications. The colors indicate the number of lipoplexes that unpack their mRNA in the cell. The histograms next to the time courses show the distribution of proteins after 30 hours without the cells that express no protein (1000000 replications). Runs where no lipoplexes unpack their mRNA in the cell and no proteins are generated are not shown.

the amount of mRNA a lipoplex contains as a specific attribute of type integer. Both solutions are possible in ML-Rules but were not possible in the tool(s) used by the authors. This led to the simplification of assuming a fixed number of mRNA per lipoplex (i.e., 350) in the model. As a result, the simulation of the protein expression in the original model shows narrow bands, depending on the number of lipoplexes that could enter the cell and unpack their mRNA (see Fig 7 left), which is not the case in the wet-lab data [75]. The authors tried to use the modeling and simulation tool SPim [76], which is based on the stochastic $\pi$ calculus [77] and allows to assign an attribute to a lipoplex that states how much mRNA it carries and unpacks this amount into the cell. However, they ran into performance issues, which made it impossible to use SPim for their study (see supplement TextS001 p. 7 from [74]).

As stated above in ML-Rules, this simplification is unnecessary, and a varying number of mRNA can be assigned to the lipoplexes modeled as compartments. The modified model can be found under http://mlrules.pages.dev/lipoplex_ext/30/h. At the beginning of the simulation, the simulation state is set to contain one cell and 200 lipoplexes containing mRNA. The amount of mRNA per lipoplex (L) is calculated based on the lipoplex size sampled from a normal distribution. The lipoplex can then move into the cell, like in the original model. Inside the cell, the lipoplex compartment L can unpack its content (?solL).

```
Cell{L{?solL}+?solC} -> Cell{?solL+?solC} @ kU;
```

The ?solL denotes the content of the lipoplex, here a population of mRNA, and ?solC is the cell's content, including other lipoplexes, mRNA, and proteins. Alternatively, we could have modeled the mRNA as an attribute of type integer. In such an implementation, the lipoplex would not be a compartment carrying the mRNA but a simple species equipped with an attribute (NmRNA) that denotes the amount of mRNA inside. The unpack rule shown above would change:

```
Cell{L+?solC} -> Cell{(L.NmRNA) mRNA+?solC} @ kU;
```

A second minor simplification is that lipoplexes can unpack their mRNA not only in the cell but also in the endosome, where the unpacked mRNA can start to deteriorate. This mechanism, called the "fully nested transfection model" by the authors (see Fig 8 in Ligon et al. [74]), is missing in the original model.

In the ML-Rules 3 model, it is realized by the two reactions:

```
Cell{E{L{?solL}+?solE}+?solC} -> Cell{E{?solL+?solE}+?solC} @ kU;
```

for the unpacking inside the endosome, and

$$E\{mRNA : m+?solE\} -> E\{?solE\} @ dM*m;$$

for the degradation of mRNA in the endosome.

Besides the simplification in the original model, some reactions, like the formation of pits, are lengthy to write down. Due to the lack of attributes or dynamic compartments, the number of lipoplexes that reside in a pit is stated in the name of the species. The original model uses ten pit species (P1—P10) denoting 1 to 10 lipoplexes inside the pit. Consequently, ten reactions need to be specified for all $P_i$ regarding the formation of pits, the endocytosis, the lysis, and the degradation of pits. This workaround only works for a low number of P-species. The growing number of reactions makes writing the model down for larger numbers increasingly harder. In ML-Rules 3, the dynamic nesting allows us to write down reactions more compactly. For example, the ten endocytosis reactions in the original model:

$$P1 -> E1$$

$$\dots$$

$$P10 -> E10$$

translate into a single rule in ML Rules 3.

$$Cell(?solC)+P\{?solP -> Cell\{E\{?solP\}+?solC\}$$

By applying the abovementioned changes, the model produces more realistic results, as can be seen by the protein expression over time in Fig 7. As stated above, the narrow bands as observed in the original model (Fig 7 left) are not observed in the wet lab experiments [74, 75]. By varying the mRNA number inside the lipoplexes, these bands in the protein expression broaden and overlap (Fig 7 right), resulting in a more realistic model behavior.

Finally, we have a look at the model's runtime performance. Therefore, we compared the runtime for executing the original model written in COPASI (version 4.41) and its ML-Rules equivalent (available at http://mlrules.pages.dev/lipoplex_orig. The original COPASI model consists of 48 reactions and 26 species compared to the ML-Rules model with its 12 rules and 8 species. This is due to the manual unrolling of the underlying nesting processes and illustrates ML-Rules' expressiveness, which also results in succinct models. We measured the runtime for 1000 replications.

As the simulation model contains events to describe the removal of external lipoplexes, only the Direct Method implementation within COPASI is able to execute the entire simulation model. This is not a limitation of the method but of the implementations. Timed events can be added relatively easily to most methods as an implementation feature. The difference in methods should be considered when interpreting the runtime measurements. For larger systems, the direct methods used generally perform worse than a more optimized logarithmic method, like the next reaction method.

We found the average runtime for a single execution to be 4.8 seconds (see Table 2). However, in two-thirds of the simulation runs, no lipoplex can unpack their mRNA into the cell,

**Table 2. Run time for different simulators.** The mRNA delivery model is executed until 30 hours (1000 replications). The ML-Rules 3 model is initialized with 201 compartments (1 cell and 200 lipoplexes) and ends with one compartment. On average, about 207 structural changes are executed during a simulation run.

| simulator | runtime [s] | | |
| --- | --- | --- | --- |
| | 20% quantile | average | 80% quantile |
| Copasi | 0.13 | 4.8 | 11.2 |
| ML-Rules 3 | 0.59 | 0.6 | 0.84 |

and consequently, no proteins are created. The 20% quantile takes 0.13 seconds, and the 80% quantile is 11.2 seconds. The equivalent simplified model in ML-Rules takes only about 0.6 seconds on average but 0.59 seconds for 20% and 0.84 seconds for the 80% quantile, respectively. Most of the runtime (87% on average) in the ML-Rules model is spent on a dynamic structure reactions, something that COPASI does not need to consider. When no mRNA is unpacked into the cell, and no proteins are created, a simulation run has as many static as dynamic structure reactions (about 200). When one lipoplex unpacks its mRNA into the cell, about 200 dynamic structure reactions happen, and about 2 million static structure reactions occur. Nevertheless, the dynamic structure reactions need about five times longer to be executed. We find similar runtimes for the adapted version of the model in ML-Rules.

## 4 Conclusion

We built ML-Rules version 3, making concise formulations of dynamic structure models run with high performance. A performance-oriented implementation of various data structures, including a partial summation tree, made this possible. We also made some changes to the language, like introducing named attributes and units of measurement. We found that this implementation outperforms the previous version of ML-Rules by two orders of magnitude. The evaluation of the simulator is based on two biological case studies. First, we used a fission yeast model that was also used in the original ML-Rules publication to show that we have a similar expressiveness but a higher performance than the previous implementation. Second, we rebuilt and extended an mRNA delivery model and showed how using dynamic compartments needs fewer simplifications than the original model that uses a static compartmental approach. The extended model matches the wet-lab data more closely, allowing a more compact notation and better performance. Finally, we built a prototypical web-based simulator using the same source code compiled to WebAssembly that can run locally on the end user machine. WebAssembly's ease of deployment and development using existing code bases and competitive performance are promising. We see further opportunities for simulation tool developers to make their software more accessible using this technology.

Our investigation has also raised some questions for future research. Currently, the network-free execution part of the model is determined by attributes explicitly defined as network-free. However, possibly a larger portion of the model could benefit from a network-free execution. To learn during simulation which part of the model to execute most efficiently in a network-based or network-free manner, possibly reinforcement learning approaches could be adapted [78].

The main simulator for ML-Rules 3 now conforms to standard SSA, with the potential exception of more complex rate expressions. It is easier to integrate with previous research on more advanced SSA variants like the partial propensity method [79] or approximate methods like advanced tau leaping [80]. Preliminary experiments with approximate tau leaping showed significant improvement for some models. However, frequently, the limiting factor is the

repeated structural conversion. Moving forward, only a subset of the simulator could be transformed if changes to the dynamic structure are localized. It would also be possible to integrate the dynamic structure more closely with the simulator, at the cost of some performance for the regular transitions. The simulator is currently applied in three different simulation studies, i.e., studying the fission and fusion of mitochondria, bone remodeling processes, and the role of endocytosis in cellular signaling.

## Acknowledgments

Fig 1 was created with BioRender.com.

## Author Contributions

**Conceptualization:** Till Köster, Tom Warnke, Adelinde Uhrmacher.

**Data curation:** Till Köster, Philipp Henning.

**Formal analysis:** Till Köster, Philipp Henning.

**Funding acquisition:** Adelinde Uhrmacher.

**Investigation:** Till Köster, Philipp Henning.

**Methodology:** Till Köster, Philipp Henning.

**Project administration:** Adelinde Uhrmacher.

**Resources:** Adelinde Uhrmacher.

**Software:** Till Köster.

**Supervision:** Adelinde Uhrmacher.

**Validation:** Till Köster, Philipp Henning.

**Visualization:** Till Köster, Philipp Henning.

**Writing – original draft:** Till Köster, Philipp Henning, Tom Warnke, Adelinde Uhrmacher.

**Writing – review & editing:** Till Köster, Philipp Henning, Adelinde Uhrmacher.

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
