## [Decision Letter · Decision Letter 0]

12 Aug 2024

PONE-D-24-26146Expressive rule-based modeling and fast simulation for dynamic compartments PLOS ONE

Dear Dr. Henning,

Thank you for submitting your manuscript to PLOS ONE. After careful consideration, we feel that it has merit but does not fully meet PLOS ONE’s publication criteria as it currently stands. Therefore, we invite you to submit a revised version of the manuscript that addresses the points raised during the review process.

We look forward to receiving your revised manuscript.

Kind regards,

Claudio Zandron

Academic Editor

PLOS ONE

Journal Requirements:

Figures 1 was created with BioRender.com. This work was funded by the Deutsche Forschungsgemeinschaft (DFG, German Research Foundation) SFB 1270 – 299150580 ELAINE and grant 225222086.

P.H. received frunding from the Deutsche Forschungsgemeinschaft (DFG, German Research Foundation, https://www.dfg.de/en) SFB 1270 – 299150580 ELAINE.

T.K. received frunding from the Deutsche Forschungsgemeinschaft (DFG, German Research Foundation, https://www.dfg.de/en) grant 225222086

The DFG did not play a role in the study design, data collection and analysis, decision to publish, or preparation of the manuscript.

**Additional Editor Comments:**

Both reviewers agree on the scientific merit of the paper. Please carefully address all minor changes/corrections they require.

Reviewers' comments:

Reviewer's Responses to Questions

**Comments to the Author**

1. Is the manuscript technically sound, and do the data support the conclusions?

Reviewer #1: Yes

Reviewer #2: Yes

2. Has the statistical analysis been performed appropriately and rigorously? 

Reviewer #1: N/A

Reviewer #2: N/A

3. Have the authors made all data underlying the findings in their manuscript fully available?

Reviewer #1: Yes

Reviewer #2: Yes

4. Is the manuscript presented in an intelligible fashion and written in standard English?

Reviewer #1: Yes

Reviewer #2: Yes

5. Review Comments to the Author

**Reviewer #1: **The authors addressed all of the points I raised in my previous

review, and have improved the paper accordingly. I am satisfied with

the revised paper.

I have only some very minor changes to suggest.

In their rebuttal, the authors clarified their use of the partial

evaluation technique. I'd recommend adding those details in the paper

as well. They already added something (around line 446) but they did

not explain *in the paper* how partial evaluation is actually used.

I was particularly interested in the fact that structure-changing

reactions are rare, yet they have a very large impact on performance

since 87% of the time is spent on those. While I am sure that this

figure (87%) may vary on the actual model, it clearly shows where the

bottleneck is.

If the authors know how frequently these rare reactions occur, I would

recommend to add this information to the paper. In the captions of

fig. 2 and 3 we can find the number of structural changes, but not

their frequency (what is the total number of fired reactions?).

line 152

BioNetgen -> BioNetGen

(there might be other occurrences)

line 194

"the containment is compartments"

(?)

**Reviewer #2:** This paper addresses the modeling and the simulation of protein interaction systems with dynamical hierarchies of compartments.

This topic is important because of the causal interactions between populations of proteins and their compartments, and back and forth.

The simulation of these systems has been acknowledged to be scientific and technological lock, mainly for scalability issues.

The approach combines a clever simulation architecture with some compilation optimization. Few test cases are provided (including pointers to the code).

One key point for simulating protein interaction systems, is the dynamical update of the table of potential events. The simulation structure encapsulates such a table for each compartment. It is presented to be applied with a reaction-based simulation approach (where rules are expanded into reactions), but the approach seems quite modular and could likely be applied with a network-free approach. This choice is motivated by the assumption that most of the events will not change the hierarchy of compartments.

To reduce the overhead of having to consider the simulation steps and the next event table for each compartment, the authors combine partial evaluation, dynamic compilation and run-time optimization.

The paper also includes a presentation of literature.

---

In my opinion, this is an important contribution to the state of the art, and it will interest the PLOS-ONE community. I thus have no doubt that it should be accepted.

Nevertheless, I think that more details should be provided about compartment fission.

This is a key limiting factor when dealing with dynamic compartments, especially when the content of the two sibling compartments is drawn by splitting randomly the content of the parent compartment, which would require many random number generation.

Such phenomena occur when the dynamics of the compartment content are driven by the well-mixed assumption.

Maybe this overhead can be avoided by using attributes, but this would deserve at least a discussion, I think.

Also, focusing on the flat models (with network-free attributes potentially), the approach focuses on the model with few diversities in the molecular entities. Indeed, models with high combinatorics in protein binding cannot be modelled accurately with such approaches.

The operational semantics of the language is only sketched.

This is a good choice to explain the intuition about the approach, but some details about the models remain difficult to understand because of this (see for instance what is the meaning to multiple occurrences of the syntactic token ?c in a rhs).

---

Please find as follows some detailed comments.

--

Line 20 [3--6]

I would recommend including references to :

Troels C. Damgaard, Espen Højsgaard, Jean Krivine,

Formal Cellular Machinery,

SASB 2012

Electronic Notes in Theoretical Computer Science,

Volume 284,

2012,

Pages 55-74,

ISSN 1571-0661,

https://doi.org/10.1016/j.entcs.2012.05.015.

and

Cardelli, L. (2005). Brane Calculi. In: Danos, V., Schachter, V. (eds) Computational Methods in Systems Biology. CMSB 2004. Lecture Notes in Computer Science(), vol 3082. Springer, Berlin, Heidelberg. https://doi.org/10.1007/978-3-540-25974-9_24

--

Line 125

An important method to speed up the dynamic update, consists in over-approximating the set of potential events. This is interesting when the over-approximation is less expensive to maintain.

The drawback is that the validity of reactions must be checked at run-time when selected randomly.

An example is the rectangular approximation in Kappa which considers the left hand side of rule connected-component by connected-component.

(40, Section 4, Page 12)

--

The dynamic updates to the table of potential events can also be optimized by exploiting the common regions among the patterns that occur in the lhs of rules.

Boutillier, B, Ehrhard T., Krivine J. Incremental Update for Graph Rewriting. ESOP 2017

--

Line 193-207

The use of an associative and commutative operator to denote a set of concurrent entities seems to be borrowed from process calculi. (eg see Ambients, BioAmabients, Brane Calculi).

--

Line 220

Even worse, the distribution of the content of the sibling compartments at the fisson of their parent compartments may be drawn stochastically. (For instance, fission of a compartment under the well-mixed assumption).

Each entity having a probability to be in one or the other compartment after fission.

This induces a computation overhead since a random decision has to be made for each entity.

eg see:

Troels C. Damgaard, Espen Hojsgaard, Jean Krivine,

Formal Cellular Machinery,

SASB 2012

Electronic Notes in Theoretical Computer Science,

Volume 284,

2012,

Pages 55-74,

ISSN 1571-0661,

https://doi.org/10.1016/j.entcs.2012.05.015.

Additional information stored in attributes, about spatial position for instance, may be used to define in which compartment each entity will be after the fission, according to the value of their attributes.

--

Line 299

Another example of named attributes are the counters in Kappa which support some arithmetic operations (inequality tests, incrementat, decrementat) and can be used in the rate of rules)

Another example of named attributes are the counters in Kappa which support some arithmetic operations (inequality tests, increment, decrement) and can be used in the rate of rules.

Boutillier B., Critescu I., Feret J.

Counters in Kappa: Semantics, Simulation, and Static analysis. ESOP 2019

--

Line 317

Discussion:

The choice between network-free and flat model simulations is usually driven by the combinatorial complexity. Network-free attibutes may help, but what about binding between proteins which are central in Kappa and BNGL.

Note also that Kappa counters could be considered as 'network-free-integers'.

--

Line 333:

A reference to unit/dimension analysis could be provided, such as:

Generalized homogeneous polynomials for efficient template-based nonlinear invariant synthesis. Kensuke Kojima, Minoru Kinoshita1, and Kohei Suenaga. In: Static Analysis Symposium (SAS 2016).

--

Line 351.

The approach seems to me to be more modular than described in the presentation. It could likely be applied to network free, flat representation, and hybrid solutions between them, depending on the nature of the model.

Compartment fission has been identified to be costly, since the distribution of the content must be drawn.

--

Line 514

Is the content of a cell is split when the cell is divided, or duplicated?

In the source code, the syntactic token ?c appears twice in the rhs. What does it mean? Is the content of the place-holder duplicated?

6. PLOS authors have the option to publish the peer review history of their article (what does this mean?). If published, this will include your full peer review and any attached files.

Reviewer #1: No

Reviewer #2: No

---

## [Author Response · Author response to Decision Letter 0]

27 Sep 2024

We thank the reviewers for their time and dedication. These detailed reviews have helped us reﬁne our paper. We have addressed the requests and made the changes detailed in the answer letter.

---

## [Decision Letter · Decision Letter 1]

15 Oct 2024

Expressive rule-based modeling and fast simulation for dynamic compartments

PONE-D-24-26146R1

Dear Dr. Henning,

We’re pleased to inform you that your manuscript has been judged scientifically suitable for publication and will be formally accepted for publication once it meets all outstanding technical requirements.

Kind regards,

Claudio Zandron

Academic Editor

PLOS ONE

Additional Editor Comments (optional):

Reviewers' comments:

Reviewer's Responses to Questions

**Comments to the Author**

1. If the authors have adequately addressed your comments raised in a previous round of review and you feel that this manuscript is now acceptable for publication, you may indicate that here to bypass the “Comments to the Author” section, enter your conflict of interest statement in the “Confidential to Editor” section, and submit your "Accept" recommendation.

Reviewer #1: All comments have been addressed

Reviewer #2: All comments have been addressed

2. Is the manuscript technically sound, and do the data support the conclusions?

Reviewer #1: Yes

Reviewer #2: Yes

3. Has the statistical analysis been performed appropriately and rigorously? 

Reviewer #1: N/A

Reviewer #2: N/A

4. Have the authors made all data underlying the findings in their manuscript fully available?

Reviewer #1: Yes

Reviewer #2: Yes

5. Is the manuscript presented in an intelligible fashion and written in standard English?

Reviewer #1: Yes

Reviewer #2: Yes

6. Review Comments to the Author

Reviewer #1: I am satisfied with the revised paper.

The authors addressed all of the points I raised in my previous

reviews.

I have no more changes to suggest.

Reviewer #2: All my comments have been considered, or discarder with proper arguments.

This paper addresses the modeling and the simulation of protein interaction systems with dynamical hierarchies of compartments.

This topic is important because of the causal interactions between populations of proteins and their compartments, and back and forth.

The simulation of these systems has been acknowledged to be scientific and technological lock, mainly for scalability issues.

The approach combines a clever simulation architecture with some compilation optimization. Few test cases are provided (including pointers to the code).

One key point for simulating protein interaction systems, is the dynamical update of the table of potential events. The simulation structure encapsulates such a table for each compartment. It is presented to be applied with a reaction-based simulation approach (where rules are expanded into reactions), but the approach seems quite modular and could likely be applied with a network-free approach. This choice is motivated by the assumption that most of the events will not change the hierarchy of compartments.

To reduce the overhead of having to consider the simulation steps and the next event table for each compartment, the authors combine partial evaluation, dynamic compilation and run-time optimization.

The paper also includes a presentation of literature.

---

In my opinion, this is an important contribution to the state of the art, and it will interest the PLOS-ONE community. I thus have no doubt that it should be accepted.

7. PLOS authors have the option to publish the peer review history of their article (what does this mean?). If published, this will include your full peer review and any attached files.

Reviewer #1: No

Reviewer #2: No

---

## [Editor Report · Acceptance letter]

18 Oct 2024

PONE-D-24-26146R1 

PLOS ONE

Dear Dr. Henning, 

I'm pleased to inform you that your manuscript has been deemed suitable for publication in PLOS ONE. Congratulations! Your manuscript is now being handed over to our production team.

Kind regards, 

on behalf of

Dr. Claudio Zandron 

Academic Editor

PLOS ONE